# Protein Kinase C (Pkc)-δ Mediates Arginine-Induced Glucagon Secretion in Pancreatic α-Cells

**DOI:** 10.3390/ijms23074003

**Published:** 2022-04-04

**Authors:** Norikiyo Honzawa, Kei Fujimoto, Masaki Kobayashi, Daisuke Kohno, Osamu Kikuchi, Hiromi Yokota-Hashimoto, Eri Wada, Yuichi Ikeuchi, Yoko Tabei, Gerald W. Dorn, Kazunori Utsunomiya, Rimei Nishimura, Tadahiro Kitamura

**Affiliations:** 1Division of Diabetes, Metabolism and Endocrinology, Department of Internal Medicine, Jikei University School of Medicine, 3-25-8 Nishishinbashi, Minato-ku, Tokyo 105-8461, Japan; mamnoon.am.m.m.h.m@gmail.com (N.H.); kazu-utsunomiya@jikei.ac.jp (K.U.); rimei@jikei.ac.jp (R.N.); 2Metabolic Signal Research Center, Institute for Molecular and Cellular Regulation, Gunma University, 3-39-15 Showa-machi, Maebashi 371-8512, Japan; masakikobayashi@gunma-u.ac.jp (M.K.); daisuke.kohno@gunma-u.ac.jp (D.K.); m10702010@gunma-u.ac.jp (O.K.); hiyokota@gunma-u.ac.jp (H.Y.-H.); wada@riem.nagoya-u.ac.jp (E.W.); m1920045@gunma-u.ac.jp (Y.I.); tabei-y@gunma-u.ac.jp (Y.T.); 3Division of Diabetes, Metabolism and Endocrinology, Department of Internal Medicine, Jikei University Daisan Hospital, 4-11-1, Izumihoncho, Komae-shi, Tokyo 201-8601, Japan; 4Center for Pharmacogenomics, Division of Cardiology, Department of Internal Medicine, Washington University School of Medicine, St. Louis, MO 63110, USA; gdorn@dom.wustl.edu

**Keywords:** pancreatic α-cell, protein kinase C-δ, glucagon, arginine

## Abstract

The pathophysiology of type 2 diabetes involves insulin and glucagon. Protein kinase C (Pkc)-δ, a serine–threonine kinase, is ubiquitously expressed and involved in regulating cell death and proliferation. However, the role of Pkcδ in regulating glucagon secretion in pancreatic α-cells remains unclear. Therefore, this study aimed to elucidate the physiological role of Pkcδ in glucagon secretion from pancreatic α-cells. Glucagon secretions were investigated in Pkcδ-knockdown InR1G9 cells and pancreatic α-cell-specific Pkcδ-knockout (αPkcδKO) mice. Knockdown of Pkcδ in the glucagon-secreting cell line InR1G9 cells reduced glucagon secretion. The basic amino acid arginine enhances glucagon secretion via voltage-dependent calcium channels (VDCC). Furthermore, we showed that arginine increased Pkcδ phosphorylation at Thr^505^, which is critical for Pkcδ activation. Interestingly, the knockdown of Pkcδ in InR1G9 cells reduced arginine-induced glucagon secretion. Moreover, arginine-induced glucagon secretions were decreased in αPkcδKO mice and islets from αPkcδKO mice. Pkcδ is essential for arginine-induced glucagon secretion in pancreatic α-cells. Therefore, this study may contribute to the elucidation of the molecular mechanism of amino acid-induced glucagon secretion and the development of novel antidiabetic drugs targeting Pkcδ and glucagon.

## 1. Introduction

The onset and progression of type 2 diabetes mellitus (T2DM) is due to relative insulin deficiency, but the contribution of impaired glucagon to T2DM is unclear. Surprisingly, it has been reported that the glucose tolerance of mice lacking pancreatic α-cells or glucagon receptors does not worsen when pancreatic β-cells are destroyed with streptozotocin [1,2,3]. This indicates that the presence of glucagon is more critical for glucose intolerance than insulin deficiency [4]. Therefore, the importance of glucagon is now being reaffirmed. 

Although still controversial, glucagon secretion from pancreatic α-cells is regulated by various intracellular signals, including a rise in Ca^2+^ influx in α-cells [5,6]. However, it is unclear whether glucose per se enhances Ca^2+^ influx in pancreatic α-cells [7,8,9,10]. Alternatively, it was also reported that Ca^2+^ influx is not necessarily involved, but cyclic adenosine monophosphate (cAMP) is involved in glucagon secretion from pancreatic α-cells [11]. Additionally, reports show that insulin affects the cAMP-PKA pathway downstream of somatostatin receptor (SSTR) 2 and regulates glucagon secretion in pancreatic α-cells [12,13]. Arginine is a basic non-essential amino acid and enhances glucagon secretion via voltage-dependent calcium channels (VDCC) [14]. It promotes the secretion of growth hormone and insulin. It is used clinically as a secretory provocation test for these hormones, but its detailed mechanism is unknown.

Protein kinase C (Pkc), a serine–threonine kinase, is ubiquitously expressed and involved in the regulation of cellular development, differentiation, death, and proliferation [15,16,17]. Pkc is divided into three sub-families, conventional, novel, and atypical, according to the difference in the activation manner. The conventional-type Pkc is activated by diacylglycerol (DAG) and Ca^2+^, while the novel-type Pkc is activated by DAG. How atypical-type Pkc is activated has not been clarified. The primary substrate for Pkc is myristoylated alanine-rich C-kinase substrate (MARCKS), a cytoskeleton component, also known as an indicator of Pkc activation [18]. MARCKS translocates from the cell membrane to cytoplasm in response to Pkc activation and regulates exocytosis [18]. Pkcδ, a novel-type Pkc, comprises the regulatory domain consisting of conserved (C) 1 and C2-like regions and the catalytic domain consisting of C3 and C4 regions. DAG binding to the Pkcδ C1 region triggers Pkcδ activation and cleavage of Pkcδ by caspase-3, which produces the Pkcδ-catalytic fragment (CF). Phosphorylation at Thr^505^ in the Pkcδ C4 region and Pkcδ translocation from the cytoplasm to the cell membrane contribute to its stabilization. 

For diabetes, hyperglycemia enhances glycolysis followed by DAG production from glyceraldehyde 3-phosphate (de novo DAG production pathway) [19]. Additionally, diabetes is often comorbid with dyslipidemia, such as free fatty acid (FFA) elevation. FFA enters cells through fatty acid transporters and is metabolized to DAG via Acyl-CoA. Therefore, Pkcδ is activated due to increased DAG levels in diabetes. Importantly, systemic Pkcδ-knockout mice had improved retinopathy by suppressing retinal vascular proliferation, as well as improved nephropathy by suppressing glomerular podocyte loss, even under diabetic conditions [20,21]. Pancreatic β-cell-specific Pkcδ kinase-negative transgenic mice had improved glucose tolerance via increased insulin secretion [22]. Therefore, Pkcδ is a therapeutic target for diabetes and diabetic complications. Alternatively, Pkcδ is expressed in pancreatic α-cells and regulates glucagon secretion [23]. Pan-Pkc activator enhanced, while Pkcδ inhibitors suppressed, glucagon secretion from isolated islets [24,25]. However, the physiological role of Pkcδ in pancreatic α-cells is unclear. Therefore, we elucidated the physiological roles of Pkcδ in pancreatic α-cells using InR1G9 cells and pancreatic α-cell-specific Pkcδ-knockout (αPkcδKO) mice.

## 2. Results

### 2.1. Knockdown of Pkcδ Decreased Glucagon Secretion in InR1G9 Cells

Pharmaceutical studies on isolated islets suggested that Pkcδ is involved in regulating glucagon secretion [25]. To confirm the physiological roles of Pkcδ in α-cells, we investigated glucagon secretion in InR1G9 cells [26,27]. Findings revealed that Pkcδ inhibitor, rottlerin, significantly decreased glucagon secretion in InR1G9 cells (Figure 1a), which is consistent with previous a report [25]. However, because rottlerin is not a specific inhibitor for Pkcδ [28], the knockdown of Pkcδ in InR1G9 cells using short interfering ribonucleic acid (siRNA) was conducted. The knockdown efficiencies by *Prkcd* #1 and #2 siRNA at the protein level were 36.0% and 46.8%, respectively (Figure 1b,c). Additionally, the expression levels of Pkcδ-CF, which is a catalytic fragment of Pkcδ and indicates its activation, were significantly reduced by these *Prkcd* siRNAs (Figure 1b,d). Because of the better knockdown efficiency, we used *Prkcd* #2 siRNA in this experiment. The knockdown efficiency by *Prkcd* #2 siRNA at the mRNA level was 49.9% in InR1G9 cells (Figure 1e). Importantly, we found that glucagon secretion was significantly reduced by Pkcδ knockdown, indicating that Pkcδ is essential for glucagon secretion in InR1G9 cells (Figure 1f).

### 2.2. Arginine-Induced Glucagon Secretion Was Decreased by Pkcδ Knockdown in InR1G9 Cells

Arginine (via activating VDCC) and 3-Isobutyl 1-methylxanthine (IBMX) (via inhibiting phosphodiesterase and increasing cAMP) are known as glucagon secretagogues. As shown in Figure 2a, both arginine and IBMX significantly increased glucagon secretion in InR1G9 cells. Arginine significantly enhanced Pkcδ phosphorylation at Thr^505^ in InR1G9 cells (Figure 2b,c), indicating that Pkcδ is activated by arginine. In these results, decreased PKCδ protein levels might be caused by enhanced PKCδ cleavage, which is also a marker for PKCδ activation. Furthermore, we investigated the localization of Pkc substrate MARCKS using green fluorescent protein (GFP)-labeled MARCKS (MARCKS-GFP). Under the basal condition, MARCKS-GFP localizing on the cell membrane was translocated to the cytoplasm by the Pkc activator, phorbol 12-myristate 13-acetate (PMA) (Figure 2d). Arginine also significantly induced cytoplasmic localization of MARCKS-GFP, which was canceled by the Pkcδ inhibitor, rottlerin (Figure 2d). These results suggested that Pkcδ is activated by arginine in InR1G9 cells. We investigated whether Pkcδ mediated arginine-induced glucagon secretion. Arginine increased glucagon secretion in InR1G9 cells, which was significantly suppressed by Pkcδ knockdown (Figure 2e). Thus, Pkcδ mediates arginine-induced glucagon secretion in InR1G9 cells.

### 2.3. Establishment of α-Cell-Specific Pkcδ-Knockout Mice

To investigate the physiological roles of Pkcδ in vivo, we generated pancreatic α-cell-specific Pkcδ-knockout mice using the Cre/LoxP system. We first crossed *Prkcd*^floxed^ [29] with *Gcg*^Cre^ mice [30]. However, because Cre recombinase expression levels in the α-cells of *Gcg*^Cre^ mice were approximately 50% (data not shown), as previously reported [31], we crossed *Prkcd*^floxed^ with *Gcg*^CreERT2^ mice, which reportedly had higher Cre expression efficiency in α-cells [32]. Because commercially available Pkcδ antibodies were not good enough for immunohistochemistry, we employed a Cre reporter mouse (*Rosa26*^tdTomato^ mouse) to test the Cre expression efficiency [33]. Native fluorescence of tdTomato allowed for direct visualization of Cre expression. Glucagon-positive and tdTomato-positive cells were mostly merged in *Gcg*^CreERT2^: *Rosa26*^tdTomato^ mice (Figure 3a). Cre expression efficiency in α-cells was 96.6% (1285 tdTomato-positive cells/1330 glucagon-positive cells) in these mice; therefore, we expect that most Pkcδ should be deleted in α-cells of *Gcg*^CreERT2^: *Prkcd*^floxed^ mice. Another advantage of *Gcg*^CreERT2^ mice was that we excluded the influence of pkcδ deletion during the embryonic stage by treating the mice with tamoxifen at the adult stage. Here, we used our homemade C-terminal glucagon antibody (rat monoclonal IgG antibody) for staining α-cells. To check the specificity of this antibody for glucagon, we performed glucagon staining together with GFP in the pancreatic sections of *Gcg*^gfp/+^ and *Gcg*^gfp/gfp^ mice [34]. Because the region including exon 2 and 3 of the glucagon gene is replaced by the gfp sequence, glucagon is not expressed in *Gcg*^gfp/gfp^ mice. Though glucagon was stained in GFP-positive α-cells of *Gcg*^gfp/+^ mice, glucagon was not detected in GFP-positive α-cells of *Gcg*^gfp/gfp^ mice (Appendix A), indicating high specificity of the glucagon antibody used in this study. We also confirmed the establishment of αPkcδKO mice at the DNA level. In this system, the recombined *Prkcd* allele (predicted band) should be detected only in Pkcδ-deleted cells. As shown in Figure 3b, the predicted and floxed bands were observed in the genomic DNA extracted from the islets of *Gcg*^CreERT2^: *Prkcd*^floxed^ mice but not in the genomic DNA extracted from the tails of these mice, showing the establishment of α-cell-specific Pkcδ-knockout mice. 

### 2.4. Arginine-Induced Glucagon Secretion Decreased in αPkcδKO Mice and the Islets from αPkcδKO Mice

We first examined the body weight, glucose tolerance, insulin sensitivity, and area of pancreatic α- and β-cells in αPkcδKO mice. There were no significant differences in body weight (Appendix A), fasting and random-fed blood glucose levels (Figure 4a,b), intraperitoneal glucose tolerance test (IPGTT) (Appendix A), and insulin tolerance test (ITT) (Appendix A) in αPkcδKO compared to the control mice. Moreover, the area of α- and β-cells and the percentage of the α-cell area against the β-cell area were comparable between the αPkcδKO and control mice (Appendix A). Furthermore, fasting and random-fed glucagon levels were similar in αPkcδKO and control mice (Figure 4c,d). However, glucagon secretion 15 min after 3 g/kg arginine intraperitoneal administration was significantly decreased in αPkcδKO mice compared to the control mice (Figure 4e). Additionally, fasting, random-fed, and arginine-induced plasma insulin levels were comparable between αPkcδKO and control mice (Appendix A) Furthermore, arginine-induced glucagon secretion was significantly lower in the islets isolated from αPkcδKO mice than the islets isolated from the control mice (Figure 4f). Altogether, although neither fasting, random-fed glucagon levels, glucose tolerance, nor insulin tolerance were unchanged, arginine-induced glucagon secretion significantly decreased in αPkcδKO mice and islets isolated from αPkcδKO mice.

## 3. Discussion

Studies have suggested that Pkcδ regulates glucagon secretion; however, these studies were mainly performed using nonspecific Pkc activators and inhibitors in isolated islets. Therefore, the specific and physiological roles of Pkcδ on glucose metabolism in vivo remain unclear. This study showed that arginine activated Pkcδ and enhanced glucagon secretion in InR1G9 cells. It also showed that arginine-induced glucagon secretion was decreased by Pkcδ knockdown in InR1G9 cells and islets isolated from αPkcδKO mice. Although arginine-induced glucagon secretion was also reduced in αPkcδKO mice, glucose and insulin tolerance remained unchanged, which is probably due to compensation mechanisms for controlling glucose metabolism. To our knowledge, this is the first report showing the physiological roles of Pkcδ in pancreatic α-cells on glucagon secretion and glucose metabolism in vivo.

Arginine enhances glucagon secretion in pancreatic α-cells, but the detailed mechanism is unclear. Arginine-induced glucagon secretion is directly inhibited by gliclazide, which suppresses adenosine triphosphate-sensitive potassium (K_ATP_) channels in α-cells [14]. Here, deletion of Pkcδ suppressed arginine-induced glucagon secretion in InR1G9 cells, αPkcδKO mice, and islets from αPkcδKO mice (Figure 2e and Figure 4e,f). Pkcδ was also reported to enhance Ca^2+^-dependent exocytosis and is related to soluble *n*-ethylmaleimide sensitive factor attachment protein receptor (SNARE) proteins’ phosphorylation [23,35]. Therefore, Pkcδ might be involved in the K_ATP_ channel–VDCC pathway and exocytosis in α-cells. In this study, although Pkcδ knockdown reduced glucagon secretion in InR1G9 cells (Figure 1f), knockout of Pkcδ did not affect glucagon secretion in αPkcδKO mice and islets from αPkcδKO mice under basal conditions (Figure 4c,d,f). This discrepancy could be caused by the paracrine effects of other islet hormones, such as insulin and somatostatin, though fasting and random-fed plasma insulin levels were unchanged in αPkcδKO (Appendix A). Indeed, insulin and somatostatin regulate glucagon secretion in α-cells by lowering cAMP [12,13]. α-cell-specific insulin receptor KO mice showed increased glucagon secretion under fasting and fed conditions [36]. Altogether, Pkcδ might be involved in the downstream of arginine signaling in α-cells, and more importantly, in arginine-induced glucagon secretion.

Because hyperglycemia increases DAG production and activates Pkcδ in various cell types, Pkcδ could be pathologically involved in diabetes progression [19]. Indeed, Pkcδ inhibition reportedly protected against the progression of diabetic nephropathy and diabetic retinopathy [20,21]. We previously reported that T2DM patients have higher plasma glucagon levels than healthy subjects, and proposed that glucagon could become a diagnostic marker and a new therapeutic target for T2DM [37,38]. This study revealed that Pkcδ is essential for arginine-induced glucagon secretion (Figure 2e and Figure 4e,f), suggesting that Pkcδ might be related to postprandial hypersecretion of glucagon in T2DM. Alternatively, Pkcδ is involved in cell proliferation and apoptosis. Furthermore, increased β-cell number and insulin content were observed in β-cell-specific Pkcδ kinase-negative transgenic mice [22]. However, in this study, the area of α-cells in αPkcδKO mice was unaltered (Appendix A), suggesting that Pkcδ deletion in α-cells does not affect α-cell proliferation and apoptosis. Thus, Pkcδ inhibition in α-cells could become a novel therapeutic target for diabetes without affecting α-cell proliferation and cell death.

This study has several limitations. First is the usage of tumorigenic cell line InR1G9. Because tumorigenic cells are oncogenic, their character may differ from native α-cells. Additionally, although InR1G9 cells secrete glucagon, they are derived from insulinoma [26]. Because the technique for isolating α-cells from islets has only been recently developed [39,40,41], we will need to reconfirm the results in InR1G9 cells using isolated α-cells in the future. The other limitation is the character of *Gcg*^CreERT2^ mice. Because *Gcg*^CreERT2^ mice possess the CreERT2-coding gene in exon 2 of the *preproglucagon* gene, they heterozygously lack the *preproglucagon* gene. Although it was reported that no significant difference was observed in glucagon secretion between *Gcg*^CreERT2/+^ and wild-type mice, the effect of heterozygous *preproglucagon* gene deletion cannot be excluded entirely [32].

In conclusion, we elucidated that Pkcδ is essential for arginine-induced glucagon secretion in pancreatic α-cells using InR1G9 cells, αPkcδKO mice, and islets from αPkcδKO mice. Thus, this study may contribute to the elucidation of the mechanism of amino acid-induced glucagon secretion and the development of novel antidiabetic drugs targeting Pkcδ and glucagon.

## 4. Materials and Methods

### 4.1. Cell Culture

Glucagon-secreting InR1G9 cells were cultured in RPMI1640 (#189-02025, Wako, Osaka, Japan) supplemented with 10% FBS (#10270-106, Life Technologies, Carlsbad, CA, USA) and 1% penicillin/streptomycin (#268-23191, Wako, Osaka, Japan). All cells were grown at 37 °C in humidified air containing 5% (*v*/*v*) CO_2_.

### 4.2. Glucagon Secretion Assay

InR1G9 cells seeded on six-well dishes or six isolated islets seeded on 24-well dishes were pre-incubated with KRB (0.13-M NaCl, 4.7-mM KCL, 1.2-mM KH_2_PO_4_, 1.2-mM MgSO_4_·7H_2_O, 10-mM HEPES, 2-μM CaCl_2_, pH 7.3) for 0.5 h. InR1G cells were incubated for 0.5 h with 15-mM arginine (#015-04613, Wako, Osaka, Japan), 100-mM IBMX (#095-03413, Wako, Osaka, Japan), or 10-µM rottlerin (#12006, CAY, Funakoshi, Japan). Isolated islets were incubated for 2 h with 15 mm arginine. Supernatants were subjected to glucagon assay using glucagon sandwich ELISA (#10-1271-01, Mercodia, Sweden). Experiments were repeated at least three times. Glucagon concentrations were normalized by total protein content using the Pierce BCA assay kit (#23227, Life Technologies, Carlsbad, CA, USA).

### 4.3. Protein Extraction and Western Blotting

Whole-cell lysates were extracted using a lysis buffer (50-mM Tris (pH 8.0), 150-mM NaCl, 10% Glycerol, 1% Triton-X100, 0.1% SDS, 5-mM EDTA, 1-mM DTT, 10- mM NaF, 1-mM Na_3_VO_4_). Equal amounts of protein samples were loaded onto SDS–PAGE gels, electrophoresed, and transferred onto nitrocellulose membranes. After blocking with non-fat milk, membranes were incubated with primary antibodies followed by the corresponding secondary horseradish peroxidase-conjugated antibodies. The signal intensity was measured using a LAS-4010 mini–Luminescent Image Analyzer (FUJIFILM, Tokyo, Japan). Pkcδ (#ab182126, abcam, Japan), phospho-Pkcδ (Thr505) (#9374, Cell Signaling Technology Japan, Tokyo, Japan), β-actin (#sc-47778, Santa Cruz Biotechnology, Dallas, TX, USA), and α-tubulin antibodies (#sc-5286, Santa Cruz Biotechnology, Dallas, TX, USA) were used.

### 4.4. RNA Extraction and Quantitative PCR

Total RNA was generated from InR1G9 cells using the RNAiso Plus kit (#9108, Takara Bio, Shiga, Japan). Additionally, RNA was reverse-transcribed into cDNA using the ImProm II Reverse Transcription System (#A3800, Promega, Madison, WI). To evaluate the mRNA expressions of Pkcδ and β-actin, quantitative real-time PCR analysis using Applied Biosystems ViiAÔ7 Real-Time PCR System (Life Technologies, Carlsbad, CA, USA) was conducted with Power SYBR Green Master Mix (#A25742, Life Technologies, Carlsbad, CA, USA). All reactions were normalized using β-actin. Primers for Pkcδ were purchased from Takara bio (#MA167545), and those for β-actin were as follows: 5′-AGCCTTCCTTCTTGGGTA-3′ (forward) and 5′-GAGCAATGATCTTGATCTTC-3′ (reverse).

### 4.5. siRNA Transfection

InR1G9 cells were transfected with scramble control (#4390843, Ambion, Life Technologies, Carlsbad, CA, USA), *Prkcd* Silencer Select siRNA #1 (#4390771, s71696, Ambion, Life Technologies, Carlsbad, CA, USA) or siRNA #2 (#4390771, s71698, Ambion, Life Technologies, Carlsbad, CA, USA) for 48 h. Additionally, transfections were performed with Lipofectamine RNAiMAX (#13778, Life Technologies, Carlsbad, CA, USA). The knockdown efficiency was evaluated 48 h after transfection, followed by Western blotting or quantitative RT-PCR analyses.

### 4.6. MARCKS Transfection and Immunocytochemistry

InR1G9 cells seeded on four-well chamber slides (# 154526, Life Technologies, Carlsbad, CA, USA) were transfected with a plasmid encoding MARCKS-GFP for 48 h using Lipofectamine 2000 transfection (#11668027, Life Technologies, Carlsbad, CA, USA), as previously described [18]. Then, InR1G9 cells were treated with 100-nM PMA (#AG-CN2-0010, Adipogen Life Sciences, San Diego, CA, USA), 15-mM arginine, or 15-mM arginine plus 10-μM rottlerin for 0.5 h. Afterwards, the cells were fixed in 4% paraformaldehyde (PFA) and the localization of MARCK-GFP was observed by a confocal laser-scanning microscope (FV1000, Olympus Life Science, Tokyo, Japan).

### 4.7. Animals and Physiological Experiments

All experimental procedures were performed in accordance with the Guide for the Care and Use of Laboratory Animals of the Science Council of Japan and approved by the Animal Experiment Committee of Gunma University. *Gcg*^GreERT2^ and *Pkcd*^floxed^ mice were mated to generate αPkcδKO mice. To induce Cre-mediated recombination, αPkcδKO mice were subcutaneously injected with 6 mg tamoxifen (2 mg, every other day) at four-weeks old and were characterized at 24-weeks old. Animal studies were performed following a normal chow diet (CE-2, CLEA Japan, Tokyo, Japan) using male mice. In this study, *Gcg*^CreERT2/+^; *Prkcd*^floxed/floxed^ and *Gcg*^+/+^; *Prkcd*^floxed/floxed^ mice served as αPkcδKO mice and controls, respectively. Arginine (pH 7.4, 3 g/kg weight) was injected intraperitoneally, and 15 min later, plasma glucagon levels were measured in αPkcδKO and control mice. Insulin (0.75 U/kg weight, Eli Lilly Japan KK, Hyogo, Japan) and D(+)-glucose (#049-31165, Wako, Osaka, Japan) were injected intraperitoneally. Blood glucose was measured using Glutest ai (#GT-1840, SANWA KAGAKU KENKYUSHO, Aichi, Japan). Plasma glucagon levels were measured by glucagon sandwich ELISA (#10-1281-01, Mercodia, Sweden). Arginine or glucose injection was performed after overnight fasting.

### 4.8. Frozen Section Preparation and Immunohistochemistry

For immunohistochemistry, αPkcδKO mice (24-week-old males) were anesthetized and perfused transcardially with ice-cold 0.05-M phosphate-buffered saline (pH 7.4) followed by 4% PFA for fixation. The pancreas was dissected and fixed in 4% PFA. Samples were frozen in OCT compound and sectioned in 5 µm-thick slices. Samples were permeabilized and blocked with 0.1% Triton-X-100/PBS/5% normal donkey serum (NDS) for 1 h. Rat monoclonal anti-glucagon (1:200, #52A1A, Immuno-Biological Laboratories, Gunma, Japan) and guinea pig polyclonal anti-insulin antibodies (1:500, #A0564, Dako, USA) were diluted in 0.1% Triton-X-100/PBS/5% NDS, then added to the sections and incubated overnight at 4 °C. After three washes with PBS, secondary antibodies (1:200, Jackson ImmunoReserch, Philadelphia, PA, USA) diluted in 0.1% Triton-X-100/PBS/5% NDS were added to the sections and incubated for 1 h at room temperature. Images were acquired using a confocal laser-scanning microscope (FV1000, Olympus Life Science, Japan).

### 4.9. PCR Detecting the Recombined Allele

Genomic DNA was extracted from the tails and the pancreatic islets of *Gcg*^CreERT2/CreERT2^; *Prkcd*^floxed/floxed^, *Gcg*^CreERT2/+^; *Prkcd*^floxed/floxed^, and *Gcg*^+/+^; *Prkcd*^floxed/floxed^ mice using REDxtract-N-Amp kit (#254-457-8, Sigma-Aldrich, St. Louis, MO, USA). PCR was performed using these DNAs as templates and the following primers: GT-1: 5′-ACCAGCGATTTGAGA AGAAGC-3′, GT-5: 5′-TCATCTGTACCTTCCACACCA-3′, GT-6: 5′-AGAACCTCCATCACGAAGAAC-3′. The recombined PCR products were loaded onto 2% agarose gel, electrophoresed, and detected using PrintgraphClassic (#WSE-5400, ATTO CORPORATION, Tokyo, Japan).

### 4.10. Pancreatic Islet Isolation

Mice were anesthetized with isoflurane and laparotomy was performed. First, the duodenum and small intestine were moved to the left side of the mouse, after which the common bile duct and papilla of Vater were checked and the proximal part (liver side) of the common bile duct was clamped. Then, a partial incision was made in the duodenum and a 27 G winged needle with a rounded tip was inserted into the common bile duct (CBD) from the duodenum. Collagenase (#C7657, Sigma-Aldrich, Japan) dissolved in medium 199 (#M0393, Sigma-Aldrich, Japan) was injected into the CBD. After confirming the swelling of the pancreas, the pancreas was excised from the rectum, stomach, duodenum, and small intestine. After that, the pancreas was added into a 50 mL tube and incubated in a water bath at 37 °C for 17 min. Then, medium 199 containing 10% FBS was added to a 50 mL tube to stop the collagenase reaction, and the mixture was shaken for 10 s. After centrifuging at 1000 rpm and 4 °C for 3 min, the supernatants were removed, then fresh medium 199 was added, and the pancreatic tissue was suspended again. The suspension was transferred to a 6 cm dish, then islets were picked up using a stereomicroscope and used in the experiment.

### 4.11. Statistical Analyses

Data were presented as mean ± SEM. Significant differences between the two groups were assessed using Student’s or Welch’s *t*-test. For multiple comparisons, we used the repeated measure one-way ANOVA with Bonferroni adjustment. A *p*-value less than 0.05 was considered significant. Statistical analyses were performed using IBM SPSS Statistics version 27 software.

## Figures and Tables

**Figure 1 ijms-23-04003-f001:**
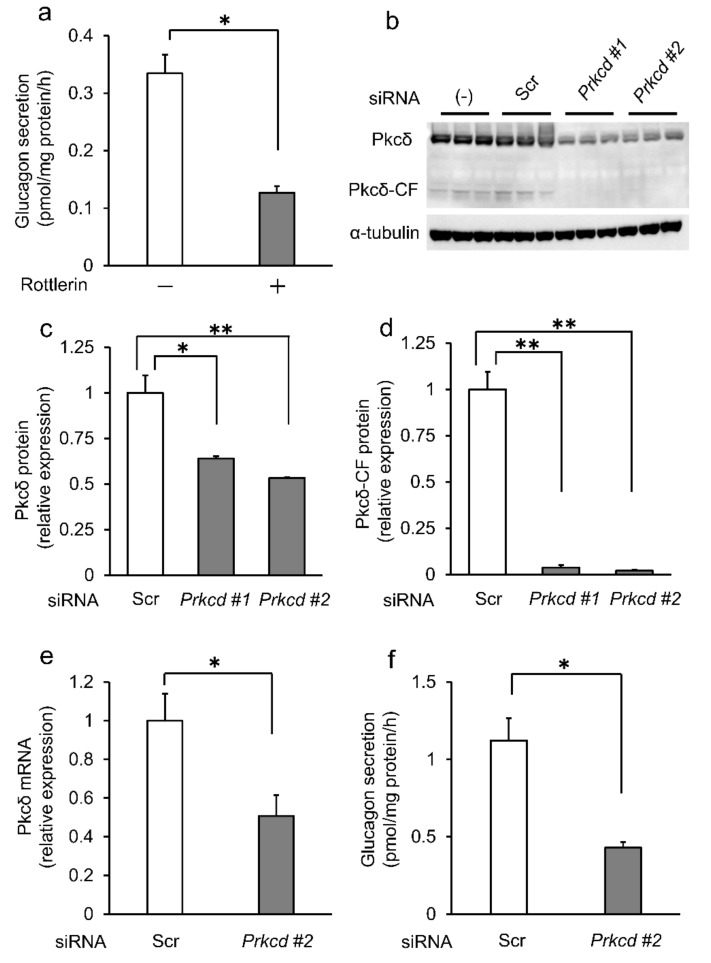
Knockdown of Pkcδ decreased glucagon secretion from InR1G9 cells. (**a**) Glucagon secretion was significantly reduced by 10-µM Pkcδ inhibitor rottlerin in InR1G9 cells. (**b**–**d**) InR1G9 cells were transfected with scrambled siRNA, *Prkcd* #1 siRNA, or *Prkcd* #2 siRNA for 48 h. Cell lysates were examined for Pkcδ and Pkcδ-CF expression by Western blotting. α-tubulin was employed as an internal control. Pkcδ expressions were reduced by around 40–50% *Prkcd* #1 siRNA and *Prkcd* #2 siRNA compared to the scrambled siRNA in InR1G9 cells. Pkcδ-CF expressions were also reduced by *Prkcd* #1 siRNA and *Prkcd* #2 siRNA in InR1G9 cells. (**e**) Pkcδ mRNA expression was significantly reduced by *Prkcd* #2 siRNA in InR1G9 cells. (**f**) Glucagon secretion was significantly reduced by *Prkcd* #2 siRNA in InR1G9 cells. We performed triplicate and three independent experiments. * *p* < 0.05, ** *p* < 0.01; *t*-test (**a**,**e**,**f**) or ANOVA (Bonferroni adjustment) (**c**,**d**). Data are expressed as means ± SEM.

**Figure 2 ijms-23-04003-f002:**
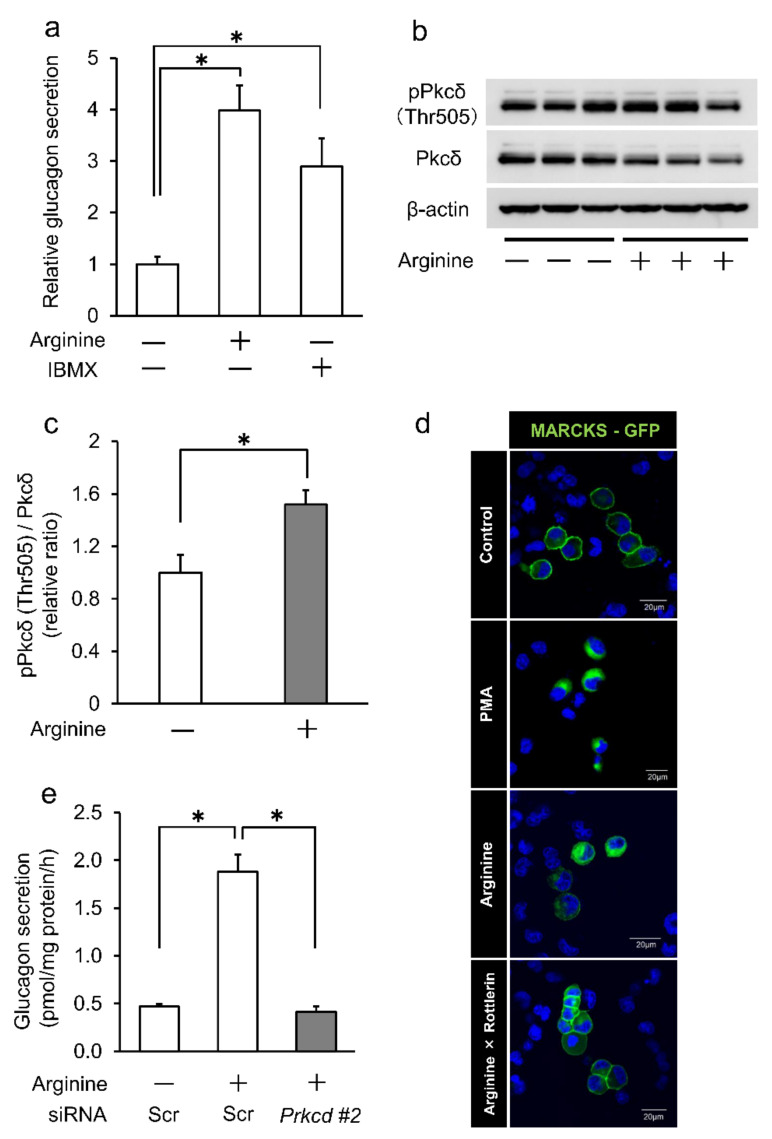
Arginine-induced glucagon secretion was decreased by Pkcδ knockdown in InR1G9 cells. (**a**) Glucagon secretion was significantly increased by 15-mM arginine and 100-mM IBMX in InR1G9 cells. (**b**–**c**) InR1G9 cells were treated with or without 15-mM arginine for 30 min; then cell lysates were examined for Pkcδ and pPkcδ (Thr505) by Western blotting. β-actin was employed as an internal control. Arginine significantly increased Pkcδ (Thr505)/Pkcδ relative ratio in InR1G9 cells. (**d**) InR1G9 cells were transfected with plasmid encoding MARCKS conjugated with GFP for 48 h followed by 100-mM PMA, 15-mM arginine, or 15-mM arginine plus 10-µM rottlerin administration. MARCKS localized on the cell membrane was translocated to the cytoplasm by PMA. Arginine also translocated MARCKS to the cytoplasm, which was canceled by rottlerin. Representative photomicrographs are shown. Scale bars represent 20 µm. (**e**) Arginine-induced glucagon secretion was significantly reduced by *Prkcd* #2 siRNA in InR1G9 cells. We performed triplicate and three independent experiments. * *p* < 0.05; *t*-test (**c**) or ANOVA (Bonferroni adjustment) (**a**,**e**). Data are expressed as means ± SEM.

**Figure 3 ijms-23-04003-f003:**
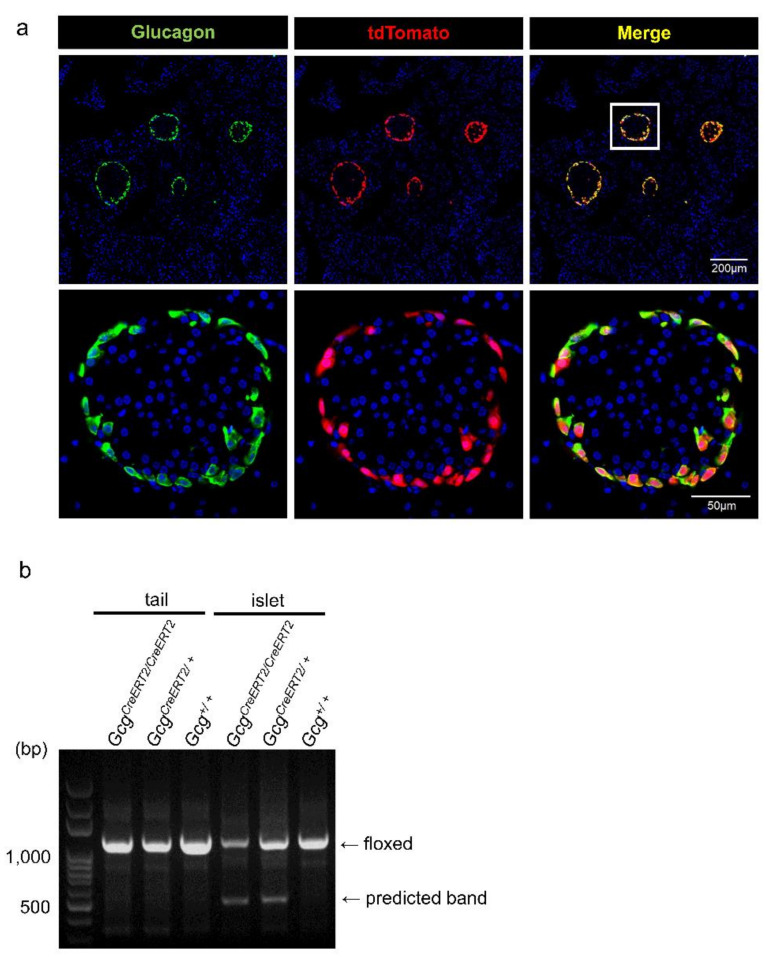
Establishment of α-cell-specific Pkcδ-knockout mice. (**a**) Pancreatic sections of *Gcg*^CreERT2^; *Rosa26*^tdTomato^ mice were subjected to immunohistochemistry with rat monoclonal anti-glucagon antibody. Native fluorescence of tdTomato is shown in red, glucagon is shown in green, and DAPI is shown in blue. Merge (yellow) indicates the colocalization of tdTomato with glucagon. Glucagon-positive and tdTomato-positive cells were mostly merged in *Gcg*^CreERT2^; *Rosa26*^tdTomato^ mice. *n* = 4, each group. Scale bars represent 200 μm (upper panels) and 50 μm (lower panels). These studies were performed under blinded analysis. (**b**) Genomic DNA was extracted from the tails and the pancreatic islets of *Gcg*^CreERT2/CreERT2^; *Prkcd*^floxed/floxed^, *Gcg*^CreERT2/+^; *Prkcd*^floxed/floxed^, and *Gcg*^+/+^; *Prkcd*^floxed/floxed^ mice. Because the region between loxP sites is excised in the tissues where Cre recombinase is expressed, the smaller PCR products (predicted bands) were detected in the islets but not in the tails of *Gcg*^CreERT2/CreERT2^; *Prkcd*^floxed/floxed^; and *Gcg*^CreERT2/+^; *Prkcd*^floxed/floxed^ mice. This result indicated that Pkcδ was deleted in the islets of αPkcδKO mice.

**Figure 4 ijms-23-04003-f004:**
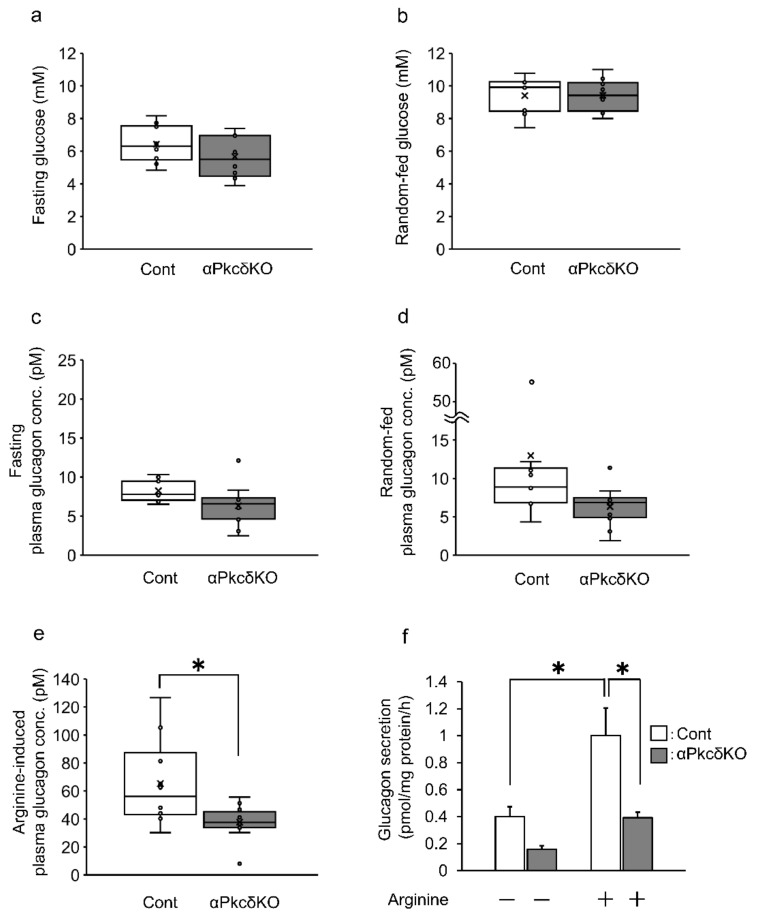
Arginine-induced glucagon secretion decreased in αPkcδKO mice and the islets from αPkcδKO mice. (**a**–**d**) Unaltered fasting (**a**) and random-fed (**b**) blood glucose levels and fasting (**c**) and random-fed (**d**) plasma glucagon levels in αPkcδKO mice. (**e**) Plasma glucagon levels 15 min after 3 g/kg arginine intraperitoneal administration was significantly lower in αPkcδKO than control mice. *n* = 10, each group. * *p* < 0.05; *t*-test. Data are expressed as a box plot. αPkcδKO indicates *Gcg^Cre^*^ERT2/+^; *Prkcd*^floxed/floxed^ mice and Cont indicates *Gcg*^+/+^; *Prkcd*^floxed/floxed^ mice. (**f**) 15-mM arginine-induced glucagon secretion was significantly lower in islets isolated from αPkcδKO mice than those from control mice. We used the islets isolated from six mice for each group (*n* = 6). We performed triplicate experiments. * *p* < 0.05; ANOVA (Bonferroni adjustment). Data are expressed as means ± SEM. αPkcδKO indicates *Gcg^Cre^*^ERT2/+^; *Prkcd*^floxed/floxed^ mice and Cont indicates *Gcg*^+/+^; *Prkcd*^floxed/floxed^ mice.

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
