# Peer review of "Protein Kinase C (Pkc)-δ Mediates Arginine-Induced Glucagon Secretion in Pancreatic α-Cells"

_ijms, 2022, doi:10.3390/ijms23074003_

Round 1
Reviewer 1 Report
The manuscript covers an interesting concept in glucagon secretion and the authors have performed an interesting set of experiments. By and large the manuscript is well written, albeit the animal study was somewhat underwhelming and some key additional measurements could have been undertaken to provide more definitive conclusion. There are a number of points to be taken into consideration:
- are there any samples stored that could measure insulin and other key metabolic/metabolic-related hormones. This would provide key information to add to the discussion.
- were any animal samples kept or processed for further gene or protein analysis to consider probing further key metabolic marker analysis?
- there does not appear to be information provided on basic animal husbandry and ethical governance of this in vivo work - please add.
- it is not clear in the legends what n=3 stands for. Please clarify the number of independent observations and replicates and include this information in the manuscript.
- please confirm that studies were only performed under basal conditions? Please expand.
- it is not clear the number of animals per group. Was it really n=3? If so, please justify.
- were immunohistochemistry studies performed under blinded analysis?
- figure legends (or methods) should clearly indicate what statistical tests were used in each analysis.
- the authors did not appear to include any effects on food intake or general behavior/activity. Please comment.
Reviewer 2 Report
In this paper, the authors sought to elucidate the physiological role of Pkcδ in glucagon secretion from pancreatic α-cells. Glucagon secretions were investigated in Pkcδ-knockdown InR1G9 cells and pancreatic α-cell-specific Pkcδ-knockout (αPkcδKO) mice. Knockdown of Pkcδ in the glucagon-secreting cell line InR1G9 cells reduced glucagon secretion. The basic amino acid arginine enhances glucagon secretion via voltage-dependent calcium channels (VDCC). Furthermore, the authors showed that arginine increased Pkcδ phosphorylation at Thr505, which is critical for Pkcδ activation. Interestingly, the knockdown of Pkcδ in InR1G9 cells reduced arginine-induced glucagon secretion. Moreover, arginine-induced glucagon secretions were decreased in αPkcδKO mice and islets from αPkcδKO mice. Pkcδ is essential for arginine-induced glucagon secretion in pancreatic α-cells. Therefore, this study may contribute to the elucidation of the molecular mechanism of amino acid–induced glucagon secretion and the development of novel antidiabetic drugs targeting Pkcδ and glucagon.
Overall, this is an interesting study, well documented and well carried out.
I congratulate the authors on this paper.
